# 3M-Diffusion: Latent Multi-Modal Diffusion for Language-Guided Molecular Structure Generation

**Huaisheng Zhu**[*1,2]**, Teng Xiao**[*1,2] **& Vasant G Honavar**[1,2,3,4]

[1]Artificial Intelligence Research Laboratory
[2]College of Information Sciences and Technology
[3]Center for Artificial Intelligence Foundations and Scientific Applications
[4]Institute for Computational and Data Sciences
The Pennsylvania State University
University Park, PA, USA
{hvz5312,tengxiao,vuh14}@psu.edu

## Abstract

Generating molecular structures with desired properties is a critical task with broad applications in drug discovery and materials design. We propose 3M-Diffusion, a novel multi-modal molecular graph generation method, to generate diverse, ideally novel molecular structures with desired properties. 3M-Diffusion encodes molecular graphs into a graph latent space which it then aligns with the text space learned by encoder-based LLMs from textual descriptions. It then reconstructs the molecular structure and atomic attributes based on the given text descriptions using the molecule decoder. It then learns a probabilistic mapping from the text space to the latent molecular graph space using a diffusion model. The results of our extensive experiments on several datasets demonstrate that 3M-Diffusion can generate high-quality, novel and diverse molecular graphs that semantically match the textual description provided. The code is available on github.

## 1 Introduction

Generating molecular structures with the desired properties is a critical task with broad applications in drug discovery and materials design (Hajduk & Greer, 2007; Mandal et al., 2009; Pyzer-Knapp et al., 2015). There is a growing interest in generative models to automate this task (You et al., 2018; Jin et al., 2018; 2020; Bjerrum & Threlfall, 2017). Kusner et al. (2017) proposed a variational autoencoder that uses parse trees for probabilistic context free grammars to encode discrete sequences that specify the Simplified Molecular-Input Line-entry System (SMILES) (Weininger, 1988) notation of molecules use the resulting parse trees to generate SMILES codes that describe molecular graphs. Edwards et al. (2021) formulated molecule retrieval from textual description as a cross-lingual retrieval task.

Recent advances in language models have led to innovative approaches to generate molecular structures that match human-friendly textual descriptions of their properties, substructures, and biochemical activity (Edwards et al., 2022; Zeng et al., 2022; Zhao et al., 2023b; Fang et al., 2023a). These approaches learn mappings from textual descriptions to molecular structures using single transformer-based cross-lingual language model for molecule generation. Specifically, with the given text description of a molecule, these approaches produce SMILES strings that encode the molecular structure.

Despite promising initial results, such approaches suffer from some important limitations: (*i*) SMILES encoding is degenerate in that the a given molecular structure can have multiple SMILE encodings. For example, the structure of ethanol can be specified using CCO, OCC or C(O)C. Although in principle a unique canonical SMILES representation can be produced

---

*Equal contribution

for a given molecular structure, the procedure used to do so relies on several arbitrary choices. Not surprisingly, molecules with identical structure may map to different SMILES strings. This makes SMILES notation less than ideal for generating molecules from their textual descriptions (Jin et al., 2018). Important chemical properties corresponding to the substructures of molecules are more straightforward to express using molecular graphs rather than linear SMILES representations (Jin et al., 2018; Du et al., 2022); (*ii*) Using language models to generate diverse and high-quality molecular structures matching a description is challenging, because the process used to decode the learned representation into the corresponding molecular structure, e.g., beam search or greedy search, often produces structures that are often too similar to each other (See Figure 2). Hence, there is an urgent need for better approaches for generating diverse, novel molecular structures from a given textual description (Brown et al., 2019). Against this background, this paper aims to answer the following research question: *How can we generate a diverse and novel collection of molecular graphs that match a given textual description?*

We introduce 3M-Diffusion, a novel multi-modal molecular diffusion method, for generating high-quality, diverse, novel molecular structures from a textual description. Inspired by text-guided image generation using a diffusion model on the latent space (Rombach et al., 2022), 3M-Diffusion is trained to generate molecular structures from the latent distribution of a molecular graph autoencoder (See Figure 1). This enables the diffusion process to focus on the high-level semantics of encodings of molecular structures.

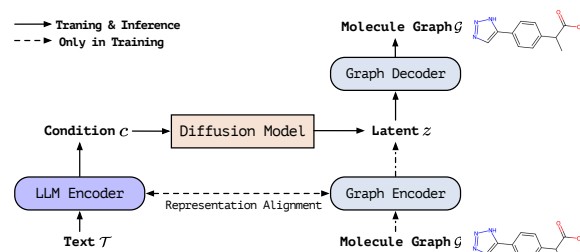

Figure 1: The overview of 3M-diffusion, with a molecular graph encoder/decoder and a latent diffusion model conditioned on a prior (an aligned LLM encoder). The details of alignment in the text/graph encoders and diffusion model are given in Section 4.

We find that this approach is particularly well-suited for discrete molecular graph modality because it relegates the challenge of modeling a discrete and diverse distribution to the autoencoder and simplifies the diffusion process by restricting it to the continuous, latent feature space. Since graph data consists of both node attributes and graph structure, which differs significantly from the sequential text data, to align the latent representations of textual descriptions and molecular graphs, 3M-diffusion employs contrastive learning on a large data set of pairs of molecular structures and their textual descriptions to pretrain molecular graph encoders and LLM text encoders. This enables 3M-diffusion to generate high-quality molecular graphs that match the given textual descriptions.

**Key Contributions.** The key contributions of this paper are as follows: (**i**) To the best of our knowledge, 3M-Diffusion offers the first multimodal diffusion approach to generating molecular structures from their textual descriptions, surpassing the limitations of state-of-the-art (SOTA) methods for this problem. (**ii**) 3M-Diffusion aligns the latent spaces of molecular graphs and textual descriptions to offer a text-molecule aligned latent diffusion model to generate higher-quality, diverse, and novel molecular structures that match the textual description provided; and (**iii**) Results of extensive experiments using four real-world text-based molecular graphs generation benchmarks show that 3M-Diffusion outperforms SOTA methods for both text-guided and unconditional molecular structure generation. In particular, 3M-Diffusion achieves 146.27% novelty, 130.04% diversity relative improvement over the SOTA method with maintained semantics in textual prompt on the PCDes dataset.

## 2 Related Work

### 2.1 Molecular Structure Generation

Existing molecular graph generation methods can be broadly grouped into two categories: text-based and graph-based models. Text-based models typically utilize the SMILES

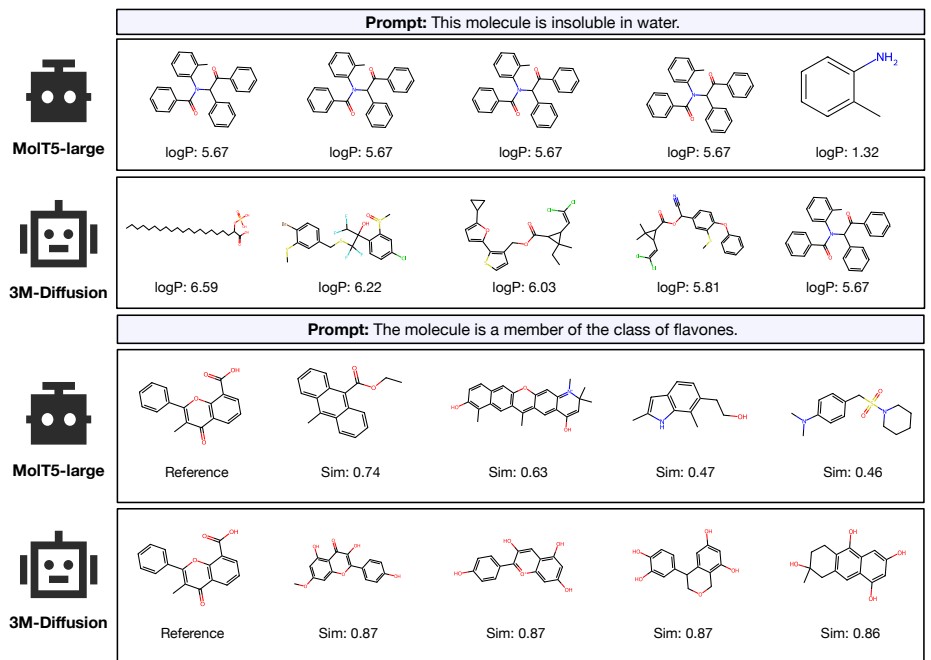

Figure 2: Qualitative comparisons to the MolT5-large of generated molecules on ChEBI-20. Compared with the SOTA method MolT5-large, our results are more diverse and novel with maintained semantics in textual prompt. More results are provided in Appendix D.4

(Weininger, 1988) or SELFIES (Krenn et al., 2022) linear strings to describe each molecule (Bjerrum & Threlfall, 2017; Gómez-Bombarelli et al., 2018; Kusner et al., 2017; Flam-Shepherd et al., 2022; Fang et al., 2023b; Grisoni, 2023). Because a given molecular graph can have several linear string representations and small changes to the string representation can result in large changes in molecular graph being described, such linear string encodings are far from ideal for learning generative models for producing molecular graph from textual descriptions. Inspired by the success of learning enhanced graph representations using graph based models (Kipf & Welling, 2016; You et al., 2020; Xiao et al.; 2024), several authors have employed deep generative models, including graph variational autoencoders (VAEs) (You et al., 2018; Jin et al., 2018; Xiao et al., 2021; Jin et al., 2020; Kong et al., 2022; Liu et al., 2018; Diamant et al., 2023), normalizing flows (Madhawa et al., 2019; Zang & Wang, 2020; Luo et al., 2021), generative adversarial networks (Guarino et al., 2017; Maziarka et al., 2020), diffusion models (Niu et al., 2020; Vignac et al., 2022; Jo et al., 2022) and autoregressive models (You et al., 2018; Xiao & Wang, 2021; Popova et al., 2019; Goyal et al., 2020) to learn distributions of molecular graphs from large databases of known graphs. However, many of these studies focus on generating molecular graphs with specific low-level properties, such as logP (the octanol-water partition coefficient) as opposed to those that match high-level textual descriptions of a broader range of molecular properties (Edwards et al., 2022), e.g., water solubility, chemical activity, etc. However, practical applications, such as drug design, call for effective approaches to generating diverse and novel molecular graphs that match a given textual description of such high-level properties.

## 2.2 Language-guided Molecule Generation

Advances in deep learning and language models have inspired a growing body of work on applications of such models to problems in the molecular sciences (Schwaller et al., 2019; Xiao et al., 2023; Toniato et al., 2021; Xiao et al., 2024; Zhao et al., 2023a; Seidl et al., 2023; Schwaller et al., 2021; Vaucher et al., 2020). Advances in text-guided generation of images, videos, and audios (Radford et al., 2021; Rombach et al., 2022; OpenAI, 2023; Yin et al., 2023), together with the flexibility offered by natural language descriptions of molecular graphs have inspired a growing body of work on text-guided generation of molecular graphs (Christofidellis et al., 2023; Edwards et al., 2022; Fang et al., 2023a; Zeng et al., 2022; Su et al., 2022). For example, MolT5 (Edwards et al., 2021; 2022) and

ChemT5 (Christofidellis et al., 2023) pre-train on a substantial volume of unlabeled language text and SMILES strings for tasks such as molecule captioning and text-based molecule generation (Christofidellis et al., 2023; Edwards et al., 2022). However, as noted above, the degenerate nature of linear SMILES string representation of molecular graphs makes it less ideal for generating molecules from their textual descriptions. To address this problem, we propose 3M-Diffusion, a novel method that aligns the latent representation of textual descriptions of molecules with that of the corresponding molecular structures, yielding a powerful approach to text-guided molecular structure generation.

## 3   Preliminaries

### 3.1   Problem Definition

We consider generative modeling of 2D molecular graphs. Each molecule is represented as a graph $\mathcal{G} = (\mathcal{V}, \mathcal{E})$, where $\mathcal{V} = \left\{ v_1, \ldots, v_{|\mathcal{V}|} \right\}$ is the set of $|\mathcal{V}|$ nodes and $\mathcal{E}$ is the set of edges. Let $\mathbf{X} = \left[ \mathbf{x}_1, \mathbf{x}_2, \cdots, \mathbf{x}_{|\mathcal{V}|} \right] \in \mathbb{R}^{[\mathcal{V}| \times D_x}$ be the node attribute matrix, where $x_i$ is the $D_x$-dimensional one-hot encoding feature vector of $v_i$, such as atomic type and chirality type. Similarly, a tensor $\mathbf{A} \in \mathbb{R}^{|\mathcal{V}| \times |\mathcal{V}| \times b}$ groups the one-hot encoding $\mathbf{e}_{ij}$ of each edge, where each entry is a distinct edge type (bond types for molecule), with the absence of an edge encoded explicitly using a designated edge type. Given the dataset $\mathcal{D} = \{ (\mathcal{G}_i, \mathcal{T}_i)_{i=1}^N \}$ where molecule $\mathcal{G}$ has specific text descriptions $\mathcal{T}$ characterizing the features of the molecule, our goal is to learn conditional generation models $p_\theta(\mathcal{G} \mid \mathcal{T})$ for generating novel, diverse, and valid molecules while also aligning with the desired features outlined in text descriptions $\mathcal{T}$. For brevity, in what follows, we omit the input subscript $i$ when it is clear from context.

### 3.2   Diffusion Models

Diffusion models (DMs) (Ho et al., 2020; Song et al., 2020) are latent variable models that represent the data $z_0$ as Markov chains $\mathbf{z}_T \cdots \mathbf{z}_0$, where intermediate variables share the same dimension. DMs involve a forward diffusion process $q\left(\mathbf{z}_{1:T} \mid \mathbf{z}_0\right) = \prod_{t=1}^T q\left(\mathbf{z}_t \mid \mathbf{z}_{t-1}\right)$ that systematically adds Gaussian noise to samples and a reverse process $p_\theta\left(\mathbf{z}_{0:T}\right) = p\left(\mathbf{z}_T\right) \prod_{t=1}^T p_\theta\left(\mathbf{z}_{t-1} \mid \mathbf{z}_t\right)$ that iteratively "denoises" samples from the Gaussian distribution to generate samples from the data distribution. A formal description of diffusion models is given in the Appendix A. To ensure training stability, we can use a simple regression objective with a denoising network $\hat{\mathbf{z}}_\theta$ with input $(\mathbf{x}_t, t)$:

$$\mathcal{L}_{\text{diff}} = \mathbb{E}_{t, \mathbf{x}_0, \epsilon} \left[ \lambda_t \left\| \hat{\mathbf{z}}_\theta \left( \sqrt{\alpha_t} \mathbf{z}_0 + \sqrt{1 - \alpha_t} \epsilon, t \right) - \mathbf{z}_0 \right\|_2^2 \right], \tag{1}$$

where $t$ is the time step, $\alpha_t \in [0, 1]$ is the noise schedule and $\epsilon \sim \mathcal{N}(\mathbf{0}, \mathbf{I})$ is Gaussian noise and $\lambda_t$ is a time-dependent weighting term. Intuitively, the denoising network is trained to denoise a noisy state, $\mathbf{z}_t = \sqrt{\alpha_t} \mathbf{z}_0 + \sqrt{1 - \alpha_t} \epsilon$, aiming to reconstruct the clean data $\mathbf{z}_0$ with Equation (1) that prioritizes specific times $t$. After training, we can draw samples with $\hat{\mathbf{z}}_\theta$ by the iterative ancestral sampling (Ho et al., 2020):

$$\mathbf{z}_{t-1} = \frac{\sqrt{\alpha_{t-1}}\left(1 - \alpha_{t|t-1}\right)}{1 - \alpha_t} \hat{\mathbf{z}}_\theta(\mathbf{x}_t, t) + \frac{\sqrt{\alpha_{t|t-1}}(1 - \alpha_{t-1})}{1 - \alpha_t} \mathbf{z}_t + \sigma(t-1, t)\epsilon, \tag{2}$$

where $\alpha_{t|t-1}$ represents $\alpha_t / \alpha_{t-1}$ and $\sigma(t-1, t) = (1 - \alpha_{t-1})\left(1 - \alpha_{t|t-1}\right) / (1 - \alpha_t)$. The sampling chain is initialized from Gaussian prior $\mathbf{z}_T \sim p\left(\mathbf{z}_T\right) = \mathcal{N}\left(\mathbf{z}_T; \mathbf{0}, \mathbf{I}\right)$.

## 4   Latent Multi-Modal Diffusion for Molecules

3M-Diffusion (See Figure 1) aims to learn a probabilistic mapping from the latent space of textual descriptions to a latent space of molecular graphs for text-guided generation of molecular graphs. However, direct mapping between these two latent representations using

a diffusion model faces a key hurdle, namely, large mismatch between the latent spaces of textual descriptions and of molecular graphs. To overcome this hurdle, 3M-Diffusion adopts a two-step approach. The first step uses contrastive learning to produce text-molecule aligned variational autoencoder that align the representation of molecular graphs with that of their textual descriptions, and a decoder that maps the latent representation back to the corresponding molecular graphs. The second step uses the text-aligned latent representation of molecular graphs to learn a conditional generative model, that maps textual descriptions to latent representations of molecular graphs. As we will see later, 3M-Diffusion can produce diverse novel molecular graphs that match the given textual descriptions.

## 4.1 Text-Molecule Aligned Variational Autoencoder

The proposed text-molecule aligned variational autonecoder comprises of three parts: the molecular graph encoder $E_g$, LLM encoder $E_t$ and molecular graph decoder $D$. To bridge the representation gap between molecular graphs and their textual descriptions, the text and molecular graph encoders are trained on a large-scale dataset of molecule-text pairs. These encoders use contrastive learning to construct well-aligned text-molecular structure encodings that align the latent representation of molecular structures by aligning it with with that of their textual descriptions. The details of this process are given below.

**Molecular Graph Encoder** aims to map each molecular graph $\mathcal{G}$ with the adjacency tensor $\mathbf{A}$ and node features matrix $\mathbf{X}$ into a lower-dimensional latent space $\mathcal{Z}$. This transformation maps the discrete space of graph structures and node attributes into a continuous latent space. Specifically, we use Graph Isomorphism Network (GIN) (Hu et al., 2019) as encoder network on the input adjacency tensor $\mathbf{A}$ and node features $\mathbf{X}$ of graph $\mathcal{G}$ to derive the mean and standard deviation of the variational marginals:

$$\boldsymbol{\mu}_g, \boldsymbol{\sigma}_g = \text{GIN}(\mathbf{A}, \mathbf{X}), \tag{3}$$

where $\boldsymbol{\mu}_g$ and $\boldsymbol{\sigma}_g$ represent the mean and standard deviations, respectively. Then, the latent graph representation $\mathbf{z}$ is sampled from a Gaussian distribution as $\mathbf{z} \sim q\left(\mathbf{z} \mid \mathbf{A}, \mathbf{X}\right) \simeq \mathcal{N}\left(\boldsymbol{\mu}_g, \boldsymbol{\sigma}_g\right)$. This graph encoding process can be expressed as $\mathbf{z} = E_g(\mathcal{G})$ signifying the outcome of the encoding operation in the input context of $\mathcal{G}$.

**LLM Encoder** maps the textual descriptions of molecular structures into a latent space that provides input to a conditional generative model of molecular graphs. This encoded information is instrumental in generating molecules that match a given textual description. To inject potentially useful scientific knowledge from the literature into $E_t$ we initialize it with encoder-based LLM, i.e., Sci-BERT (Beltagy et al., 2019), which is pretrained on the text of scientific publications.

**Aligning the representation of molecular graphs with that of their textual descriptions** is crucial for establishing a shared representation for effective text-guided molecular graph generation. Graph data encodes both node attributes and graph structure which differs significantly from the sequential and contextual nature of text data. We aim to align the molecular representation from the molecule graph encoder $E_g$ with text representation from the LLM encoder $E_t$ using contrastive learning. Consider a molecular graph $\mathcal{G}$ and its textual description $\mathcal{T}$. To align these two encoders and ensure a cohesive latent space, we employ the following contrastive learning loss over a batch $\mathcal{B} \in \mathcal{D}$ of large-scale dataset containing molecule-text pairs to pretrain encoders:

$$\mathcal{L}_{\text{con}} = -\frac{1}{|\mathcal{B}|} \sum_{(\mathcal{G}_i, \mathcal{T}_i) \in \mathcal{B}} \log \frac{\exp\left(\cos\left(\mathbf{z}_i, \mathbf{c}_i\right) / \tau\right)}{\sum_{j=1}^{|\mathcal{B}|} \exp\left(\cos\left(\mathbf{z}_i, \mathbf{c}_j\right) / \tau\right)}, \tag{4}$$

where $\mathbf{z}_i = E_g(\mathcal{G}_i)$, $\mathbf{c}_i = E_t(\mathcal{T}_i)$ and $\mathbf{c}_j = E_t(\mathcal{T}_j)$ are representations of corresponding molecule graph and texts, respectively. $\cos(\cdot, \cdot)$ denotes cosine similarity and $\tau$ is the temperature hyperparameter. The encoders are pretrained on molecule-text (300K) pairs from PubChem Liu et al. (2023). As we will see, this process yields a well-aligned molecule-text space that can support the generation of diverse and novel molecular structures from their textual descriptions.

**Molecular Graph Decoder** learns to use latent representation $\mathbf{z}$ to generate the corresponding molecular structure. In this work, we employ the HierVAE (Jin et al., 2020) decoder. Note however that our approach can be combined with other graph decoders (Li et al., 2018; Kong et al., 2022; Grover et al., 2019). Specifically, the molecular graph decoder generates a molecular structure $\mathcal{G}$ using a graph representation $\mathbf{z}$ from the encoder. Given the molecular graph $\mathcal{G}$, we minimize the negative Evidence Lower Bound (ELBO) (Jin et al., 2020) to train both the encoder and the decoder:

$$\mathcal{L}_{\text{elbo}} = -\mathbb{E}_{q(\mathbf{z}|\mathbf{A},\mathbf{X})}[p(\mathcal{G} \mid \hat{\mathcal{G}})] + \alpha \text{KL}[q(\mathbf{z} \mid \mathcal{G})\|p(\mathbf{z})]. \tag{5}$$

Here $\hat{\mathcal{G}} = D(\mathbf{z})$, in which $D$ denotes the decoder parameterized by HierVAE (Jin et al., 2020). $\text{KL}[q(\cdot)\|p(\cdot)]$ is the Kullback-Leibler (KL) divergence between $q(\cdot)$ and $p(\cdot)$. We follow Jin et al. (2020) and set the weight of KL loss $\alpha = 0.1$. Here, $p(\mathbf{z}) = \mathcal{N}(\mathbf{z} \mid \mathbf{0}, \mathbf{I})$ is the isotropic multivariate Gaussian prior, where $\mathbf{I}$ is the identity matrix. We optimize this first expectation term concurrently using the reparameterization trick (Kingma & Welling, 2013).

## 4.2 Multi-modal Molecule Latent Diffusion

We next introduce latent diffusion for conditional molecular graph generation, a method that leverages the aligned molecule-text latent space of encoder-decoder described above, enabling us to preserve the crucial molecular properties and aligning the textual description of $\mathcal{G}$. Specifically, we learn a probabilistic mapping from the text to the text aligned molecular structure latent space, and thus generate molecular structures that match the conditional text inputs.

The denoising network $\hat{\mathbf{z}}_\theta$ focuses on generating latent graph representation $\mathbf{z}$ conditioned on the text representation $\mathbf{c}$ from the LLM encoder $E_t$. The objective to be optimized is:

$$\mathcal{L}_{\text{diff}} = \mathbb{E}_{t,\mathbf{x}_0,\epsilon} \left[ \lambda_t \left\| \hat{\mathbf{z}}_\theta \left( \sqrt{\alpha_t}\mathbf{z} + \sqrt{1 - \alpha_t}\epsilon, t, \mathbf{c} \right) - \mathbf{z} \right\|_2^2 \right], \tag{6}$$

where $\lambda_t$ are time-dependent weights and we use MLPs for the network $\hat{\mathbf{z}}_\theta$. Specifically, we concatenate $\mathbf{c}$ to graph latent representation $\mathbf{z}_t$ for timestep $t$ as the input of $\hat{\mathbf{z}}_\theta$. The denoising network is trained to denoise a noisy latent, $\mathbf{z}_t = \sqrt{\alpha_t}\mathbf{z} + \sqrt{1 - \alpha_t}\epsilon$ to the clean $\mathbf{z}$. $\mathbf{c}$ is the text representation from the fixed encoder $E_t$ pretrained by Equation (4).

Additionally, to improve sample quality we take advantage of classifier-free guidance (Ho & Salimans, 2022), to jointly train an unconditional network, $\hat{\mathbf{z}}_\theta(\mathbf{z}_t, t)$, and a conditional network, $\hat{\mathbf{z}}_\theta(\mathbf{z}_t, t, \mathbf{c})$. Conditioning information is randomly dropped with a probability of $p = 0.1$ during training. When conditioning information is dropped, we replace the embedded source text with its embedding.

## 4.3 Training and Inference

**Training**. We proceed to describe the training procedure for 3M-Diffusion. While the learning objectives for graph encoder-decoder and multi-modal latent diffusion have already been specified by Equations (5) and (6), it is unclear whether the two components should be trained sequentially. Previous research on latent diffusion models for image generation indicates that a two-stage training strategy often results in superior performance (Rombach et al., 2022). Hence, we adopt a similar two-stage strategy. The first stage consists of training an encoder-decoder pair using Equation (5), and the second stage consists of training a multi-modal molecule latent diffusion with the fixed encoder-decoder from the first stage.

**Inference**. During inference, we use a weight $w$ in computing the prediction as follows (Ho & Salimans, 2022):

$$\tilde{\mathbf{z}}_t = w\hat{\mathbf{z}}_\theta(\mathbf{z}_t, t, \mathbf{c}) + (1 - w)\hat{\mathbf{z}}_\theta(\mathbf{z}_t, t), \tag{7}$$

where $\tilde{\mathbf{z}}_t$ represents the newly predicted results, incorporating both conditional and unconditional information. Setting $w = 1.0$ corresponds to the conditional diffusion model, while setting $w > 1.0$ enhances the impact of the conditioning information. A smaller value of $w$ contributes to more diverse samples. The new sampling equation is:

$$\mathbf{z}_{t-1} = \frac{\sqrt{\alpha_{t-1}}\left(1 - \alpha_{t|t-1}\right)}{1 - \alpha_t}\tilde{\mathbf{z}}_t + \frac{\sqrt{\alpha_{t|t-1}}\left(1 - \alpha_{t-1}\right)}{1 - \alpha_t}\mathbf{z}_t + \sigma(t - 1, t)\epsilon, \tag{8}$$

Table 1: Quantitative comparison of conditional generation on the PCDes and MoMu. 3M-Diffusion outperforms other SOTA methods in terms of novelty, diversity, and validity metrics by a large margin, while maintaining a good similarity metric.

| # Metrics | PCDes | | | | MoMu | | | |
|---|---|---|---|---|---|---|---|---|
| | Similarity (%) | Novelty (%) | Diversity (%) | Validity (%) | Similarity (%) | Novelty (%) | Diversity (%) | Validity (%) |
| MolT5-small | 64.84 | 24.91 | 9.67 | 73.96 | 16.64 | 97.49 | 29.95 | 60.19 |
| MolT5-base | 71.71 | 25.85 | 10.50 | 81.92 | 19.76 | 97.78 | 29.98 | 68.84 |
| MolT5-large | **88.37** | 20.15 | 9.49 | 96.48 | **25.07** | 97.47 | 30.33 | 90.40 |
| ChemT5-small | 86.27 | 23.28 | 13.17 | 93.73 | 23.25 | 96.97 | 30.04 | 88.45 |
| ChemT5-base | 85.01 | 25.55 | 14.08 | 92.93 | 23.40 | 97.65 | 30.07 | 87.61 |
| Mol-Instruction | 60.86 | 35.60 | 24.57 | 79.19 | 14.89 | 97.52 | 30.17 | 68.32 |
| **3M-Diffusion** | 81.57 | **63.66** | **32.39** | **100.0** | 24.62 | **98.16** | **37.65** | **100.0** |

Table 2: Quantitative comparison of conditional generation on the ChEBI-20 and PubChem. 3M-Diffusion outperforms other SOTA methods in terms of novelty, diversity, and validity metrics by a large margin, while maintaining a good similarity metric.

| # Methods | ChEBI-20 | | | | PubChem | | | |
|---|---|---|---|---|---|---|---|---|
| | Similarity (%) | Novelty (%) | Diversity (%) | Validity (%) | Similarity (%) | Novelty (%) | Diversity (%) | Validity (%) |
| MolT5-small | 73.32 | 31.43 | 17.22 | 78.27 | 68.36 | 20.63 | 9.32 | 78.86 |
| MolT5-base | 80.75 | 32.83 | 17.66 | 84.63 | 73.85 | 21.86 | 9.89 | 79.88 |
| MolT5-large | **96.88** | 12.92 | 11.20 | 98.06 | **91.57** | 20.85 | 9.84 | 95.18 |
| ChemT5-small | 96.22 | 13.94 | 13.50 | 96.74 | 89.32 | 20.89 | 13.10 | 93.47 |
| ChemT5-base | 95.48 | 15.12 | 13.91 | 97.15 | 89.42 | 22.40 | 13.98 | 92.43 |
| Mol-Instruction | 65.75 | 32.01 | 26.50 | 77.91 | 23.40 | 37.37 | 27.97 | 71.10 |
| **3M-Diffusion** | 87.09 | **55.36** | **34.03** | **100** | 87.05 | **64.41** | **33.44** | **100** |

where the sampling chain is initialized from Gaussian prior $\mathbf{z}_T \sim p(z_T) = \mathcal{N}(\mathbf{z}_T; \mathbf{0}, \mathbf{I})$. The detailed training and inference algorithms of 3M-diffusion are provided in Appendix B.

## 5 Experiments and Results

### 5.1 Experimental Setup

**Datasets.** We use PubChem (Liu et al., 2023), ChEBI-20 (Edwards et al., 2021), PCDes (Zeng et al., 2022) and Momu (Su et al., 2022) datasets. Following the evaluation methodology adopted with datasets (ZINC250K (Irwin et al., 2012) and QM9 (Blum & Reymond, 2009; Rupp et al., 2012)) used in previous studies of molecular structure generation, we limit our training, validation, and test data to molecules with fewer than 30 atoms. Statistics of datasets are shown in Table 5 and detailed descriptions are given in the Appendix C.1.

**Evaluation.** We evaluate the model's performance on two tasks: text-guided and unconditional molecule generation. Following previous works on text-guided molecule generation, we compare our 3M-Diffusion the SOTA baselines (Edwards et al., 2022; Christofidellis et al., 2023; Fang et al., 2023a). Text-guided molecule generation measures the model's capacity to generate chemically valid, structurally diverse, novel molecules that match the textual description provided. As in previous work (Chen et al., 2021; Fu et al., 2021; Wang et al., 2022), we adopt the following metrics for this task: *Similarity*, *Novelty*, *Diversity*, *Validity*. We call a predicted molecular structure, specifically one that is chemically valid ($g'$), *qualified* with the ground truth graph $g$ if $f(g, g') > 0.5$, where $f$ measures the MACCS (Durant et al., 2002) cosine similarity between the two. If the $f(g, g')$ similarity between $g'$ and $g$ is smaller than a threshold (0.8 in this paper), we call this predicted molecule $g$ *novel*. Specifically, (**i**) *Similarity*: the percentage of qualified molecules out of 20K molecules; (**ii**) *Novelty*: the percentage of novel molecules out of all qualified molecules; (**iii**) *Diversity*: the average pairwise distance $1 - f(\cdot, \cdot)$ between all qualified molecules. (**iv**) *Validity*: the percentage of generated molecules that are chemically valid out of all molecules.

We report the results for unconditional molecule generation tasks using the GuacaMol benchmarks (Brown et al., 2019) to assess the model's ability to learn a generative model of molecular structures that can be used to generate realistic, diverse molecules. Specifically, *Uniqueness* gauges the ratio of unique molecules among the generated ones. *Novelty* measures the models' capacity to generate molecules not present in the training set. *KL Divergence* assesses the proximity between the distributions of various physicochemical properties in the training set and the generated molecules. *Fréchet ChemNet Distance* (FCD) computes the proximity of two sets of molecules based on their hidden representations in

Table 3: Quantitative comparison of unconditional generation. Results of Uniq, KL Div and FCD on ChEBI-20 and PubChem, which refer to Uniqueness, KL Divergence and Fréchet ChemNet Distance, respectively. A higher number indicates a better generation quality.

| # Methods | ChEBI-20 | | | | | PubChem | | | | |
|---|---|---|---|---|---|---|---|---|---|---|
| | Uniq (%) | Novelty (%) | KL Div (%) | FCD (%) | Validity (%) | Uniq (%) | Novelty (%) | KL Div (%) | FCD (%) | Validity (%) |
| CharRNN | 72.46 | 11.57 | 95.21 | 75.95 | 98.21 | 63.28 | 23.47 | 90.72 | 76.02 | 94.09 |
| VAE | 57.57 | 47.88 | 95.47 | 74.19 | 63.84 | 44.45 | 42.47 | 91.67 | 55.56 | 94.10 |
| AAE | 1.23 | 1.23 | 38.47 | 0.06 | 1.35 | 2.94 | 3.21 | 39.33 | 0.08 | 1.97 |
| LatentGAN | 66.93 | 57.52 | 94.38 | 76.65 | 73.02 | 52.00 | 50.36 | 91.38 | 57.38 | 53.62 |
| BwR | 22.09 | 21.97 | 50.59 | 0.26 | 22.66 | 82.35 | **82.34** | 45.53 | 0.11 | 87.73 |
| HierVAE | 82.17 | 72.83 | 93.39 | 64.32 | **100.0** | 75.33 | 72.44 | 89.05 | 50.04 | **100.0** |
| PS-VAE | 76.09 | **74.55** | 83.16 | 32.44 | **100.0** | 66.97 | 66.52 | 83.41 | 14.41 | **100.0** |
| **3M-Diffusion** | **83.04** | 70.80 | **96.29** | 77.83 | **100.0** | **85.42** | 81.20 | **92.67** | 58.27 | **100.0** |

ChemNet (Preuer et al., 2018). Each metric is normalized to a range of 0 to 1, with a higher value indicating better performance.

**Baselines.** For text guided molecule generation, we compare 3M-Diffusion with the following baselines: MolT5 (Edwards et al., 2022), ChemT5 (Christofidellis et al., 2023) and Mol-Instruction (Fang et al., 2023a). We consider seven representative baselines for unconditional molecule generation based on Variational Autoencoder, including CharRNN (Segler et al., 2018), VAE (Kingma & Welling, 2013), AAE (Makhzani et al., 2015), LatentGAN (Prykhodko et al., 2019) BwR (Diamant et al., 2023), HierVAE (Jin et al., 2020) and PS-VAE (Kong et al., 2022). The details of the baselines are given in Appendix C.1.

**Setup.** We randomly initialize the model parameters and use the Adam optimizer to train the model. HierVAE is used as our variational autoencoder for all of the following experiments. For baselines of unconditional generation that did not report results in these datasets, we reproduce the results using the official code made available by the authors. To ensure a fair comparison across all methods for unconditional generation, we select the best hyperparameter configuration solely based on the loss on the training set. We evaluate the performance of baselines for text-guided molecule generation using the pretrained parameters made available by the authors of the respective studies.

## 5.2 Text-guided Molecule Generation

Table 1 and 2 report *Similarity*, *Novelty*, *Diversity*, and *Validity* for text-guided molecule generation on four datasets. Each metric is computed as an average over 5 structures generated from a given textual description. We see that 3M-Diffusion is competitive with SOTA baselines with respect to *Similarity* and outperforms them substantially with respect to Diversity and Novelty. Specifically, 3M-Diffusion achieves 146.27% (Novelty of PCDes), 130.04% (Diversity of PCDes) relative

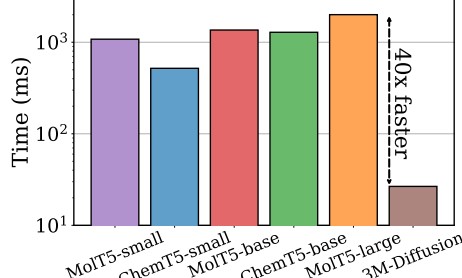

Figure 3: Inference time comparison for conditional molecule generation on ChEBI-20.

improvement over the second-best SOTA baseline while matching the textual prompt. We conclude that while the transformer based models that are pretrained on a large dataset excel in producing molecular structures that match the textual descriptions provided, 3M-Diffusion outperforms them with respect to the diversity and novelty of the molecular structures without sacrificing the match between the textual descriptions provided as input and the molecular structures generated as output. Also, as Figure 3 shows, 3M-Diffusion enjoys a substantial advantage in terms of its speed over transformer-based methods (See Appendix D.1 for details).

## 5.3 Unconditional Molecule Generation

Table 3 shows the results of unconditional molecule generation on PubChem and ChEBI-20. 3M-Diffusion consistently outperforms or matches the performance of SOTA baselines across all four metrics on both datasets, suggesting that it is able to generate realistic, diverse, and novel molecules without overfitting the training data. Furthermore, 3M-Diffusion consistently outperforms HierVAE when both methods use the same decoder.

Table 4: Results of ablation study with different model designs of 3M-diffusion for Pubchem.

| # Methods | Similarity (%) | Novelty (%) | Diversity (%) | Validity (%) |
|---|---|---|---|---|
| **(i)** Without two-stage training | 77.74 | 67.03 | 31.05 | **100.0** |
| **(ii)** Without representation alignment | 86.21 | 62.81 | 32.78 | **100.0** |
| **(iii)** Without two-stage training & representation alignment | 10.24 | **77.89** | 27.86 | **100.0** |
| **3M-Diffusion** | **87.05** | 64.41 | **33.44** | **100.0** |

This underscores the clear advantage of 3M-Diffusion in generating a diverse and novel molecular graphs (See Figure 10 for 50 samples from the learned distribution).

## 5.4 Ablation Studies

We conduct ablation studies to assess the impact of different model designs, by joint training of the diffusion model and Variational Autoencoder ("without two-stage training") and training the model without aligning the representation space of graphs and text using $\mathcal{L}_{con}$ in Equation (4) ( "without representation alignment"). Additionally, we perform experiments without *both* two-stage training and representation alignment ("without two-stage training & representation alignment") in Table 4. We find that joint training the aligned encoder-decoder alongside Multi-modal Molecule Latent Diffusion achieves comparable or superior performance relative to 3M-Diffusion in terms of diversity and novelty, but at the expense of reduced quality of match between the molecular structures produced and the textual descriptions provided. We find that training the model without the aligned encoder-decoder results in substantial drop in performance relative to 3M-Diffusion, underscoring the importance of aligning the representation of textual descriptions of molecules with that of molecular structures. Ablation of both two-stage training and representation alignment components leads to a substantial drop in performance (Similarity of 10.24), rendering the resulting model unsuitable for real-world applications. The full 3M-Diffusion model (last row) achieves the best performance with respect to all of the metrics. This confirms the importance of two-step training and representation alignment in training a model that can generate diverse, novel set of molecular structures from their textual descriptions.

## 5.5 Qualitative Analysis and Case Study

**Property Conditional Molecule Generation.** In traditional molecule generation models, the generation process is often conditioned on specific molecular properties such as logP (Kusner et al., 2017) and QED. logP assesses a molecule's solubility, while QED evaluates its drug likeness (Bickerton et al., 2012). Previous methods have explored ways to optimize such models further in order to generate molecules with desired properties. In contrast, 3M-Diffusion utilizes textual prompts for conditional molecule generation (See Figure 2). In particular, we generated 10 samples for each prompt and subsequently select the top 5 samples based on the value of the desired property. For MolT5-large, we adjust the temperature of the transformer in an attempt to encourage diversity while ensuring that invalid samples are controlled during the sampling process. We applied the same methodology used in our previous experiments to maintain consistency across the comparative analysis, as shown in Figure 2. When compared to the MolT5-large model, we find that 3M-Diffusion excels in generating a diverse range of molecules that also score better with respect to the desired properties, as evidenced by higher logP values Figure 2. This underscores our 3M-Diffusion's ability to produce molecules that are not only novel and diverse but also match the textual description provided. This unique approach distinguishes 3M-Diffusion from prior methods for molecular generation, presenting a fresh and promising perspective on molecular design. Additional experimental details and insights regarding prompts related to solubility and drug likeness are provided in Appendix D.2.

**Qualitative Comparison.** In Figure 2, we introduce more specific prompts related to molecular categories, such as "The molecule is a member of the class of flavones." These prompts bias the model in favor of generating useful drug candidates. Specifically, we choose a reference molecule from the training dataset that matches the given textual prompt. Subsequently, we generate ten additional molecules and select the top five molecules that exhibit the highest structural similarity (calculated by Rascal algorithm (Raymond et al., 2002)) to the reference molecule. These selected molecules are shown in Figure 2.Comparison of 3M-

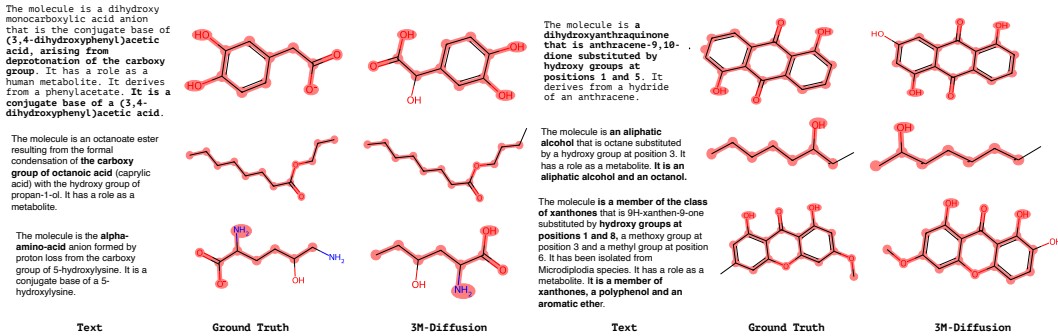

Figure 4: Molecules generated conditionally on input text by 3M-Diffusion on ChEBI-20.

Diffusion with MolT5-large shows that 3M-Diffusion generates more diverse molecules that exhibit a higher degree of structural similarity with the reference molecule while producing novel molecular structures. Additional examples of prompts are provided in Appendix D.3.

**Case Study.** In Figure 4, we choose a set of examples from the ChEBI-20 dataset to visually illustrate the structural similarity between the molecules generated by 3M-Diffusion and the ground truth structures corresponding to the given textual descriptions. Specifically, we show the largest substructure shared by the generated and ground truth molecules (using RDkit (Landrum, 2010)). 3M-Diffusion demonstrates the capability to generate molecules that include the relevant substructure, as illustrated in Figure 4. Simultaneously, it displays the flexibility to generate molecules with novel substructures, thereby striking a balance between generating structures that are similar to those represented in the training data and suggesting novel structures that match the textual description provided. We include additional results in Appendix D.4.

# 6    Conclusion

Generating molecules with desired properties is a critical task with broad applications in drug discovery and materials design. Such applications call for methods that produce diverse, and ideally novel, molecules with the desired properties. We have introduced 3M-Diffusion, a novel multi-modal molecular graph generation method, to address this challenge. 3M-Diffusion first encodes molecular graphs into a graph latent space aligned with text descriptions. It then reconstructs the molecular graph and atomic attributes based on the given text descriptions using the molecule decoder. 3M-Diffusion learns a probabilistic mapping from the text space to the latent molecular graph space using the latent diffusion model. The results of extensive experiments show that 3M-Diffusion can generate high-quality, novel and diverse molecular structures that semantically match the textual description provided. Ablation studies underscore the critical role of aligning the latent representation of text with that of molecular graphs in 3M-Diffusion.

# Acknowledgments

We thank the anonymous reviewers and Prof. Dane Morgan of University of Wisconsin-Madison for suggestions that have helped us improve the paper. This work was supported in part by grants from the National Science Foundation (2020243, 2226025) to Vasant Honavar and the National Center for Advancing Translational Sciences, and the National Institutes of Health (UL1 TR002014).

# Impact Statements

This paper advances machine learning approaches to text-guided generation of molecular structures, with potential application in drug design and material discovery. The resulting molecules, if successfully synthesized and experimentally characterized in the lab, could provide new drugs for combating diseases and for new materials for wide ranging applications.

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

## A Diffusion Models

We present a formal description of diffusion (Ho et al., 2020; Song et al., 2020). Diffusion models are latent variable models that represent the data $\mathbf{z}_0$ as Markov chains $\mathbf{z}_T \cdots \mathbf{z}_0$, where intermediate variables share the same dimension. DMs involve a forward diffusion process $q\left(\mathbf{z}_{1:T} \mid \mathbf{z}_0\right) = \prod_{t=1}^{T} q\left(\mathbf{z}_t \mid \mathbf{z}_{t-1}\right)$ and a reverse process $p_\theta\left(\mathbf{z}_{0:T}\right) = p\left(\mathbf{z}_T\right) \prod_{t=1}^{T} p_\theta\left(\mathbf{z}_{t-1} \mid \mathbf{z}_t\right)$. The forward process starting from $\mathbf{z}_0$ can be written as:

$$q\left(\mathbf{z}_t \mid \boldsymbol{x}_0\right) := \int q\left(\mathbf{z}_{1:t} \mid \boldsymbol{x}_0\right) d\mathbf{z}_{1:(t-1)} = \mathcal{N}\left(\mathbf{z}_t; \sqrt{\alpha_t}\boldsymbol{x}_0, (1-\alpha_t)\boldsymbol{I}\right), \tag{9}$$

where the hyperparameter $\alpha_{1:T}$ determines the magnitude of noise added at each timestep $t$. The values of $\alpha_{1:T}$ are selected to ensure that samples $\mathbf{z}_T$ converge to standard Gaussians, i.e., $q\left(\mathbf{z}_T\right) \approx \mathcal{N}(0, \boldsymbol{I})$. This forward process $q$ is usually predefined without trainable parameters.

The generation process of DMs is defined as learning a parameterized reverse denoising process, which incrementally denoises the noisy variables $\mathbf{z}_{T:1}$ to approximate clean data $x_0$ in the target data distribution:

$$p_\theta\left(\mathbf{z}_{t-1} \mid \mathbf{z}_t\right) = \mathcal{N}\left(\mathbf{z}_{t-1}; \boldsymbol{\mu}_\theta\left(\mathbf{z}_t, t-1, t\right), \sigma^2(t-1, t)\boldsymbol{I}\right), \tag{10}$$

where the initial distribution $p(\mathbf{z}_T)$ is defined as $\mathcal{N}(0, \boldsymbol{I})$. The means $\mu_\theta$ typically are neural networks such as U-Nets for images or Transformers for text, and the variances $\sigma^2(t-1, t)$ typically are also predefined. As latent variable models, the forward process $q(\mathbf{z}_{1:T}|\mathbf{z}_0)$ can be seen as a fixed posterior. The reverse process $p_\theta(\mathbf{z}_{0:T})$ is trained to maximize the variational lower bound of the likelihood of the data:

$$\mathcal{L}_{\text{vlb}} = \mathbb{E}_{q(\mathbf{z}_{1:T}|\mathbf{z}_0)} \left[\log \frac{q\left(\mathbf{z}_T \mid \mathbf{z}_0\right)}{p_\theta\left(\mathbf{z}_T\right)} + \sum_{t=2}^{T} \log \frac{q\left(\mathbf{z}_{t-1} \mid \mathbf{z}_0, \mathbf{z}_t\right)}{p_\theta\left(\mathbf{z}_{t-1} \mid \mathbf{z}_t\right)} - \log p_\theta\left(\mathbf{z}_0 \mid \mathbf{z}_1\right)\right] \tag{11}$$

To ensure training stability, a simple surrogate objective up to irrelevant constant terms is introduced (Song et al., 2020; Ho et al., 2020; Nichol & Dhariwal, 2021):

$$\mathcal{L}_{\text{diff}} = \mathbb{E}_{t,\mathbf{z}_0,\epsilon} \left[\lambda_t \left\|\hat{\mathbf{z}}_\theta\left(\sqrt{\alpha_t}\mathbf{z}_0 + \sqrt{1-\alpha_t}\epsilon, t\right) - \mathbf{z}_0\right\|_2^2\right], \tag{12}$$

where $\mathbf{z}_t$ is used to denote $\sqrt{\alpha_t}\mathbf{z}_0 + \sqrt{1-\alpha_t}\epsilon$ for simplicity. Intuitively, we train a neural network $\hat{\mathbf{z}}_\theta(\mathbf{z}_t, t)$ to approximate the original data given some noisy latent and the timestep through Equation (12), $\hat{\mathbf{z}}_\theta\left(\mathbf{z}_t, t\right) \approx \mathbf{z}$. With a trained denoising network, we define the mean and variance function in Equation (10) following the previous work (Ho et al., 2020) as:

$$\boldsymbol{\mu}_\theta\left(\mathbf{z}_t, t-1, t\right) = \frac{\sqrt{\alpha_{t-1}}\left(1-\alpha_{t|t-1}\right)}{1-\alpha_t}\hat{\mathbf{z}}_\theta(\mathbf{z}_t, t) + \frac{\sqrt{\alpha_{t|t-1}}\left(1-\alpha_{t-1}\right)}{1-\alpha_t}\mathbf{z}_t,$$
$$\sigma^2(t-1, t) = \frac{\left(1-\alpha_{t-1}\right)\left(1-\alpha_{t|t-1}\right)}{1-\alpha_t}, \tag{13}$$

where $\lambda_t$ is the time-dependent weight and $\alpha_{t|t-1} = \alpha_t / \alpha_{t-1}$. For generation, we employ the standard DDPM sampler, also referred to as the ancestral sampler (Ho et al., 2020). The process involves sampling initial noise $\mathbf{z}_T \sim \mathcal{N}(\mathbf{0}, \mathbf{I})$ and iteratively applying the update rule:

$$\mathbf{z}_{t-1} = \frac{\sqrt{\alpha_{t-1}}\left(1-\alpha_{t|t-1}\right)}{1-\alpha_t}\hat{\mathbf{z}}_\theta(\mathbf{z}_t, t) + \frac{\sqrt{\alpha_{t|t-1}}\left(1-\alpha_{t-1}\right)}{1-\alpha_t}\mathbf{z}_t + \sigma(t-1, t)\epsilon, \tag{14}$$

where $\epsilon \sim \mathcal{N}(\mathbf{0}, \mathbf{I})$. We use $T = 50$ for the sampling timesteps.

---

**Algorithm 1** Training Algorithm of 3M-Diffusion

---

1: **Input:** Pairs of Molecule Graph data $\mathcal{G} = (\mathcal{V}, \mathcal{E})$ and Textual Description $\mathcal{T}$
2: **Initial:** Graph Encoder network $E_g$, LLM Encoder network $E_t$ and Decoder network $D$, denoising network $\hat{\mathbf{z}}_\theta$
3: **Pretraining Stage: Aligned Text and Graph Encoder Training**
4: **while** $E_g$ and $E_t$ have not converged **do**
5: $\quad \mathcal{L}_{\text{con}} = -\frac{1}{|\mathcal{B}|} \sum_{(\mathcal{G}_i, \mathcal{T}_i) \in \mathcal{B}} \log \frac{\exp(\cos(\mathbf{z}_i, \mathbf{c}_i)/\tau)}{\sum_{j=1}^{|\mathcal{B}|} \exp(\cos(\mathbf{z}_i, \mathbf{c}_j)/\tau)}$
6: $\quad E_g, E_t \leftarrow \text{optimizer}(\mathcal{L}_{\text{con}})$
7: **end while**
8: **First Stage: Autoencoder Training**
9: Initialize $E_g$ from **Pretraining Stage**
10: **while** $E_g$ and $D$ have not converged **do**
11: $\quad \mathbf{z} \leftarrow E_g(\mathcal{G})$
12: $\quad \mathcal{G} \leftarrow D(\mathbf{z})$                                               ▶ Decoding
13: $\quad \mathcal{L}_{\text{elbo}} = -\mathbb{E}_{q(\mathbf{z}|\mathbf{A},\mathbf{X})}[p(\mathcal{G} \mid \hat{\mathcal{G}})] + \alpha \text{KL}[q(\mathbf{z} \mid \mathcal{G}) \| p(\mathbf{z})]$
14: $\quad E_g, D \leftarrow \text{optimizer}(\mathcal{L}_{\text{elbo}})$
15: **end while**
16: **Second Stage: Multi-modal Molecule Latent Diffusion Training**
17: Fix Graph Encoder $E_g$ from **Stage 1** and LLM Encoder $E_t$ from **Pretraining Stage**
18: **while** $\mathbf{z}_\theta$ have not converged **do**
19: $\quad \mathbf{z} \leftarrow E_g(\mathcal{G})$
20: $\quad t \sim \mathbf{U}(0, T), \boldsymbol{\epsilon} \sim \mathcal{N}(\mathbf{0}, \boldsymbol{I})$
21: $\quad \mathcal{L}_{\text{diff}} = \mathbb{E}_{t,\mathbf{x}_0,\boldsymbol{\epsilon}} \left[ \lambda_t \left\| \hat{\mathbf{z}}_\theta \left( \sqrt{\alpha_t}\mathbf{z}_0 + \sqrt{1-\alpha_t}\boldsymbol{\epsilon}, t \right) - \mathbf{z}_0 \right\|_2^2 \right]$
22: $\quad \theta \leftarrow \text{optimizer}(\mathcal{L}_{\text{diff}})$
23: **end while**
24: **return** $E_g, D$

---

---

**Algorithm 2** Sampling Algorithm of 3M-Diffusion

---

1: **Input:** decoder network $D$, denoising network $\hat{\mathbf{z}}_\theta$
2: $\mathbf{z}_T \sim \mathcal{N}(\mathbf{0}, \boldsymbol{I})$
3: **for** $t$ in $T, T-1, \cdots, 1$ **do**
4: $\quad \boldsymbol{\epsilon} \sim \mathcal{N}(\mathbf{0}, \boldsymbol{I})$                                         ▶ Latent Denoising Loop
5: $\quad \tilde{\mathbf{z}}_t = w\hat{\mathbf{z}}_\theta(\mathbf{z}_t, t, \mathbf{c}) + (1-w)\hat{\mathbf{z}}_\theta(\mathbf{z}_t, t),$
6: $\quad \mathbf{z}_{t-1} = \frac{\sqrt{\alpha_{t-1}}(1-\alpha_{t|t-1})}{1-\alpha_t}\tilde{\mathbf{z}}_t + \frac{\sqrt{\alpha_{t|t-1}}(1-\alpha_{t-1})}{1-\alpha_t}\mathbf{z}_t + \sigma(t-1, t)\boldsymbol{\epsilon}$
7: **end for**
8: $\mathcal{G} \sim p(\mathcal{G}|\mathbf{z}_0)$                                               ▶ Decoding
9: **return** $\mathbf{x}, \mathbf{h}$

---

## B   Training and Sampling Algorithm

The complete training algorithm is depicted in Algorithm 1. Firstly, the process begins by training the graph and LLM encoders using contrastive learning. Following that, the Variational Autoencoder is trained to facilitate the reconstruction of molecule graphs. Lastly, the Multi-modal Molecule Latent Diffusion Model is employed to train the denoising network for the sampling process. This structured approach forms the core of our training procedure. Moreover, we present the sampling method of 3M-Diffusion in Algorithm 2.

Table 5: Statistics of all datasets.

| Dataset | #Training | #Validation | #Test |
|---------|-----------|-------------|-------|
| ChEBI-20 | 15,409 | 1,971 | 1,965 |
| PubChem | 6,912 | 571 | 1,162 |
| PCDes | 7,474 | 1,051 | 2,136 |
| MoMu | 7,474 | 1,051 | 4,554 |

## C  Experimental Details

### C.1  Datasets Details and Statistics

The statistics of datasets, including their the number of samples for training, validation and test sets are given in Table 5.

**ChEBI-20** (Edwards et al., 2021): This dataset is generated by leveraging PubChem (Kim et al., 2016) and Chemical Entities of Biological Interest (ChEBI) (Hastings et al., 2016). It compiles ChEBI annotations of compounds extracted from PubChem, comprising molecule-description pairs. To ensure less noise and more informative molecule descriptions, it includes samples with descriptions exceeding 20 words.

**PubChem** (Liu et al., 2023): This dataset comprises 324k molecule-text pairs gathered from the PubChem website. As the dataset contains numerous uninformative texts like "The molecule is a peptide," it curates a high-quality subset of 15k pairs with text longer than 19 words for downstream tasks. This refined subset is then randomly split into training, validation, and test sets. The remaining dataset, which is more noisy, is utilized for pretraining. To prevent data leakage, we exclude samples that appear in the testing set of other datasets from the pretraining dataset.

**PCDes** (Zeng et al., 2022): This dataset compiles molecule-text pairs sourced from Pub-Chem (Kim et al., 2016), encompassing names, SMILES notations, and accompanying paragraphs providing property descriptions for each molecule.

**MoMu** (Su et al., 2022): This dataset comprises paired molecule graph-text data, with the textual information for each molecule extracted from the SCI paper dataset (Lo et al., 2019).

For all datasets, inspired by previous datasets (ZINC250K (Irwin et al., 2012) and QM9 (Blum & Reymond, 2009; Rupp et al., 2012)) on molecule generation, we preserve molecules with fewer than 30 atoms, creating new training, validation, and test sets from their original counterparts.

### C.2  Baselines

We firstly introduce the baselines used for text-guided molecule generation in this section.

**MolT5** (Edwards et al., 2022): MolT5 addresses the difficult problem of cross-domain generation by linking natural language and chemistry, tackling tasks such as text-conditional molecule generation and molecule captioning. It presents several T5-based LMs (Raffel et al., 2020) by training on the SMILES-to-text and text-to-SMILES translations.

**ChemT5** (Christofidellis et al., 2023): ChemT5 introduces a versatile multi-task model for seamless translation between textual and chemical domains. This innovative approach leverages transfer learning in the chemical domain, with a particular emphasis on addressing cross-domain tasks that involve both chemistry and natural language.

Then, we will also include seven baselines for unconditional generation to verify the effectiveness of our model.

**CharRNN** (Segler et al., 2018): CharRNN models a distribution over the next token based on the previously generated tokens. We train this model by maximizing the log-likelihood of the training data, which is represented as SMILES strings.

**VAE** (Kingma & Welling, 2013): Variational Autoencoder (VAE) consists of two neural networks: an encoder and a decoder. The encoder infers a mapping from high-dimensional data to a lower-dimensional space, while the decoder maps this lower-dimensional representation back to the original high-dimensional space. It optimizes reconstruction loss and regularization term in a form of Kullback-Leibler divergence. We use SMILES as input and output representations as discussed in (Polykovskiy et al., 2020).

**AAE** (Makhzani et al., 2015): Adversarial Variational Autoencoder (AAE) replaces the Kullback-Leibler divergence in VAE with an adversarial objective. An auxiliary discriminator network is trained to distinguish between samples from a prior distribution and the model's latent codes. Similar to VAE, we use SMILES as input and output representations as shown in (Polykovskiy et al., 2020).

**LatentGAN** (Prykhodko et al., 2019): The Latent Vector Based Generative Adversarial Network (LatentGAN) integrates an autoencoder with a generative adversarial network. It first pretrains an autoencoder to map SMILES structures to latent vectors. Subsequently, a generative adversarial network is trained to generate latent vectors for the pre-trained decoder.

**BwR** (Diamant et al., 2023): BwR aims to decrease the time/space complexity and output space of graph generative models, demonstrating notable performance improvements in generating high-quality outputs on biological and chemical datasets.

**HierVAE** (Jin et al., 2020): HierVAE employs a hierarchical encoder-decoder architecture to generate molecular graphs, leveraging structural motifs as fundamental building blocks.

**PS-VAE** (Kong et al., 2022): PS-VAE automatically identifies regularities within molecules, extracting them as principal subgraphs. Utilizing these extracted principal subgraphs, it proceeds to generate molecules in two distinct phases.

Note that we use the code for CharRNN, VAE, AAE, LatentGAN by the Molecular Sets (MOSES) Benchmark Polykovskiy et al. (2020).

### C.3   Setup and Hyper-parameter settings

We use official implementation publicly released by the authors of the baselines or implementation from Pytorch. Note that we use the decoder of HierVAE for our model. We run experiments on a machine with a NVIDIA A100 GPU with 80GB of GPU memory. In all experiments, we use the Adam optimizer (Kingma & Ba, 2014). In particular, for our 3M-diffusion. we set the dimension of latent representation as 24, the number of layers for the denosing networks (MLP) as 4, $\alpha$ as 0.1, $T$ as 100 for training and 50 for sampling process and the probability $p = 0.8$ for text-guided molecule generation. For unconditional generation, we set $T$ as 10 or 5 for the sampling process.

## D   More Experimental Results

### D.1   Average Inference Time Comparison

In this section, we compare the average inference times of 3M-Diffusion, MolT5, and ChemT5 by generating 1,000 samples, as shown in Figure 3. The average time to generate each molecule is calculated and presented. We observe that 3M-Diffusion achieves a speed 45 times faster than baseline methods, confirming the efficiency of our proposed model. Our proposed model not only generates molecules more quickly than previous approaches but also maintains a high level of semantic alignment with textual descriptions.

### D.2   Property Conditional Molecule Generation

In this section, we provide more prompt examples and more experiment details for Property Conditional Molecule Generation. We generate 10 samples for each prompt and subsequently select the top 5 samples based on the desired property values, as shown in Figure 2, 5,  6 and  7. For instance, when using the prompt "This molecule is not like a drug," we

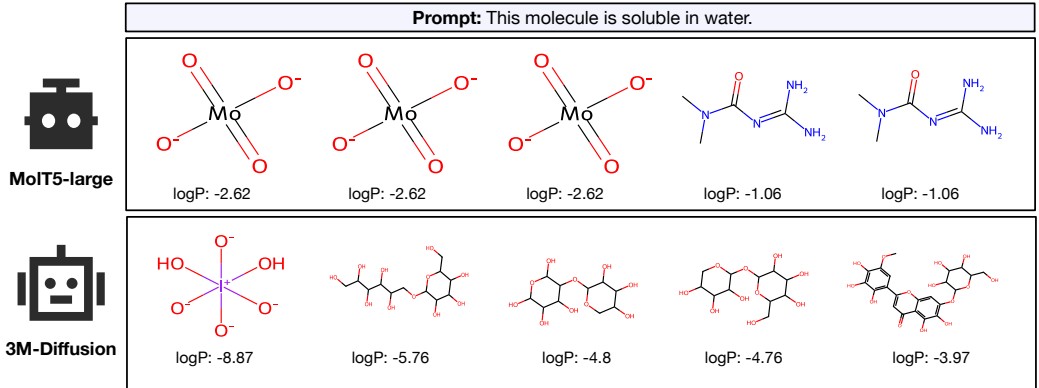

Figure 5: Qualitative comparisons to the MolT5-large in terms of generated molecules on CheBI-20. Compared with the SOTA method MolT5-large, our generated results are more diverse and novel with maintained semantics in textual prompt.

select the top 5 molecules with the smallest QED values, as indicated. Conversely, for the prompt "This molecule is like a drug," we opt for the top 5 molecules with the largest QED values. This same approach is applied to solubility, where we select molecules based on logP values. Specifically, we adjust temperature setting within the MolT5 transformer model to promote the generation of diverse molecules. We carefully selected an appropriate temperature threshold that struck a balance between diversity and maintaining the validity of the generated molecules. Moreover, when dealing with MolT5, it is worth noting that it may occasionally generate multiple unconnected molecules within a single SMILES representation. In such cases, we select the molecule among these unconnected ones that best aligns with the top desired attribute. From all figures, it becomes evident that our model excels in generating molecules with improved desired properties when compared to MolT5-large.

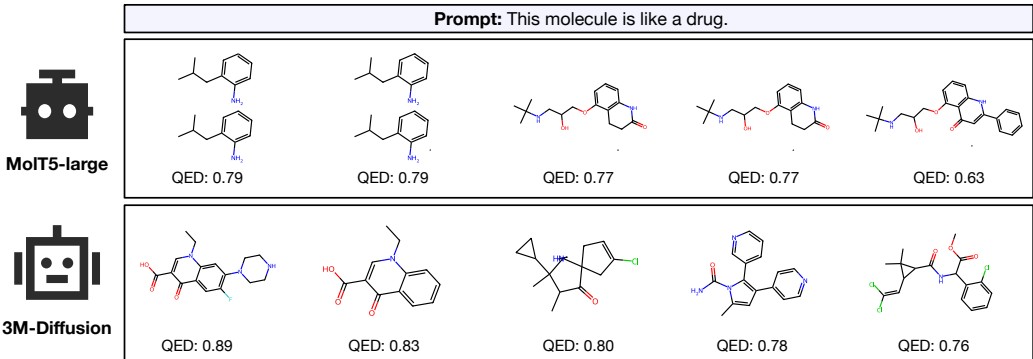

Figure 6: Qualitative comparisons to the MolT5-large in terms of generated molecules on CheBI-20. Compared with the SOTA method MolT5-large, our generated results are more diverse and novel with maintained semantics in textual prompt.

## D.3 Qualitative Comparison

In this section, we introduce an additional prompt example: "The molecule is an antho-cyanidin cation." This prompt is used to generate molecules belonging to specific categories. We have the same observation that our model excels in generating molecules that not only closely resemble the reference molecule but also encompass some diverse and novel structures.

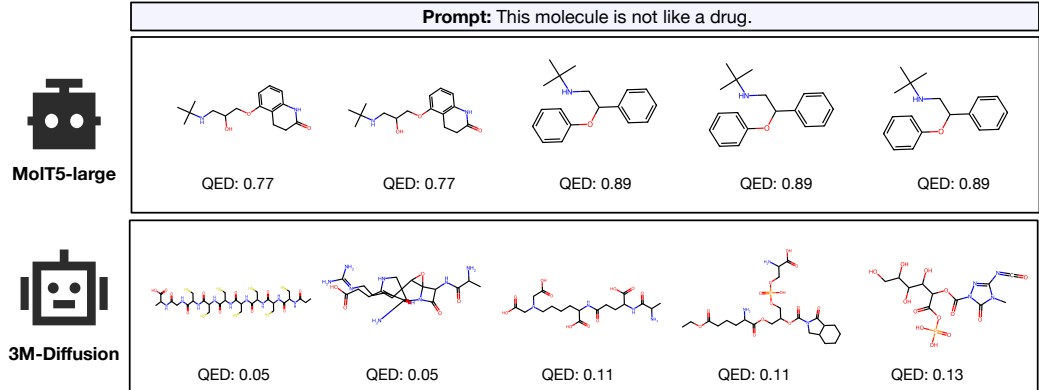

Figure 7: Qualitative comparisons to the MolT5-large in terms of generated molecules on CheBI-20. Compared with the SOTA method MolT5-large, our generated results are more diverse and novel with maintained semantics in textual prompt.

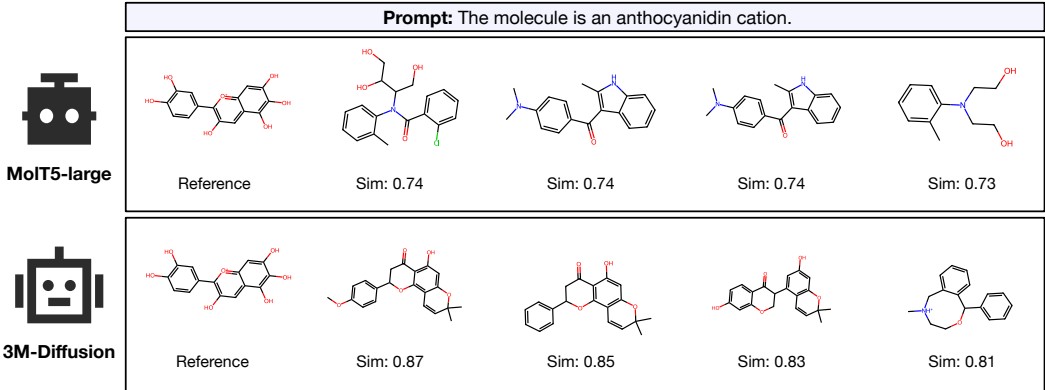

Figure 8: Qualitative comparisons to the MolT5-large in terms of generated molecules on CheBI-20. Compared with the SOTA method MolT5-large, our generated results are more diverse and novel with maintained semantics in textual prompt.

### D.4 Case Study

In this section, we choose 20 more examples from the ChEBI-20 dataset to visually illustrate the structural similarity between the molecules generated by 3M-Diffusion and the ground truth molecules corresponding to the given text descriptions. All results are shown in Figure 4. We also observe that our model is proficient at generating molecules that share significant common structural elements, while simultaneously obtaining diverse and innovative molecular structures.

### D.5 Unconditional Molecule Generation

As shown in Figure 10, our sampled molecules present rich variety and structural complexity. This demonstrates our model's ability to generate diverse, novel, and realistic molecules.

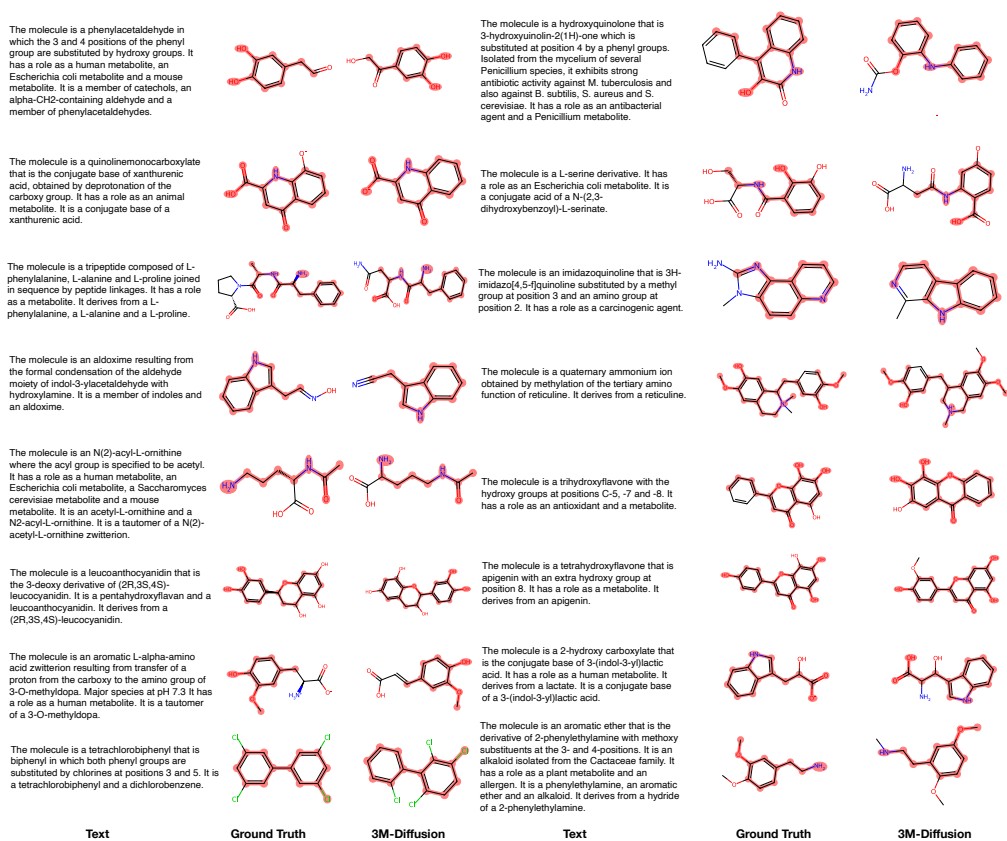

Figure 9: More molecules generated conditionally on input text by 3M-Diffusion on the ChEBI-20 dataset.

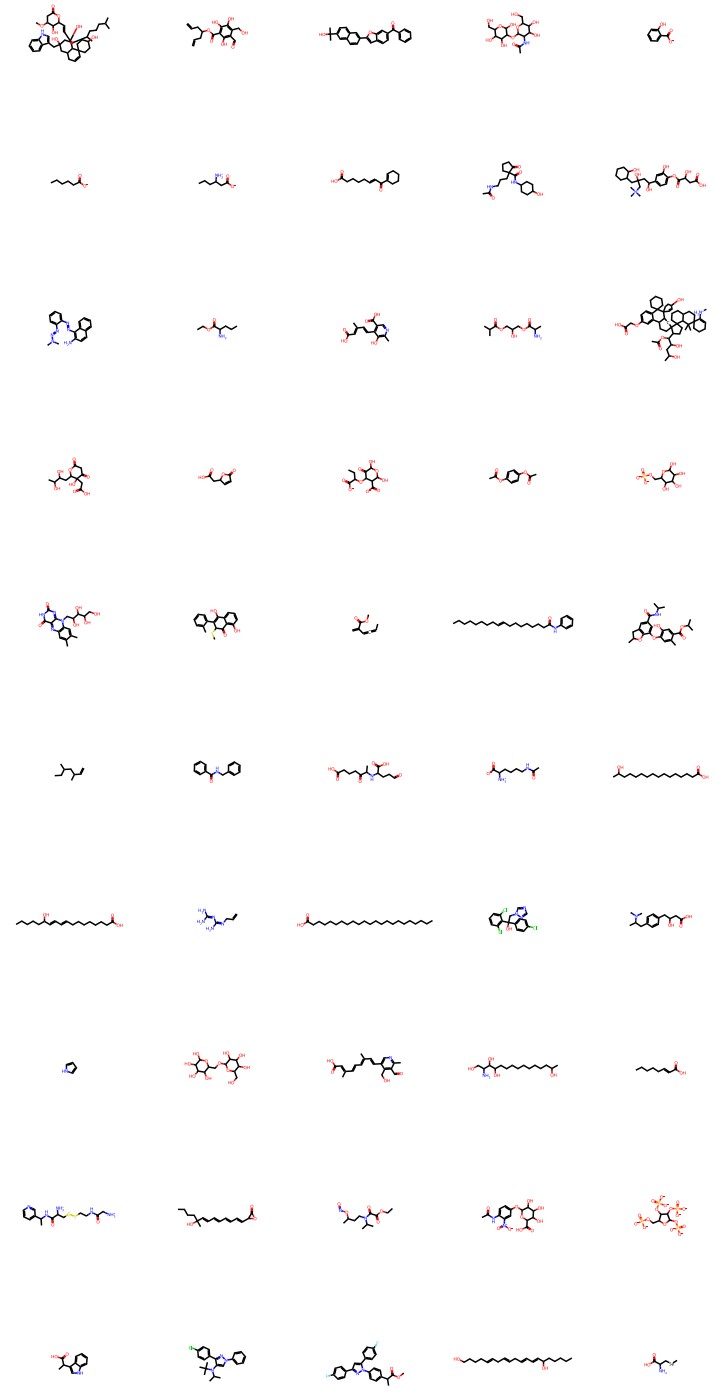

Figure 10: 50 molecules sampled from the prior distribution $\mathcal{N}(\mathbf{0}, \mathbf{I})$.

