# OpenReview forum: "3M-Diffusion: Latent Multi-Modal Diffusion for Language-Guided Molecular Structure Generation"
_colmweb.org/COLM/2024/Conference — COLM_

### Official Review · Reviewer_xhrH · 2024-05-10

**Rating:** 4
**Confidence:** 4
**Ethics Flag:** 1

**Summary:**

This paper proposes a method for text-guided molecule generation. The method leverages a latent diffusion model to generate latent representations of molecular graphs. Further, a variational graph autoencoder is employed to decode a molecular graph from the latent representation. For text-guided generation, an LLM encoder is trained to align with the graph encoder by cross-modal contrastive learning. The output of the LLM encoder is also incorporated as a condition to control the latent diffusion model.

**Questions To Authors:**

cf. above.

**Reasons To Accept:**

- The method is overall sound and novel. Applying diffusion models for text-guided molecule generation is an important application.
- The literature review is comprehensive and effectively positions this work within the context of existing studies.

**Reasons To Reject:**

* In Introduction, the authors promised results on the GuacaMol benchmarks, yet these results are absent from the subsequent sections.
* The evaluation metrics in Table 1 and Table 2 are insufficient. Why are the evaluation metrics from MolT5 (*e.g.*, BLEU, Exact, Levenshtein, MACCS) missing in this work?
* **Unconditional molecule generation (Table 5)**
  * Since no textual condition is used, the comparison should include baselines in [1].
  * Table 5 should be moved to the main body of the paper.
* Table 1 and Table 2 have the same caption, yet the contents are different. The authors should specify the distinct contents of each table to avoid confusion.

[1] Molecular Sets (MOSES): A Benchmarking Platform for Molecular Generation Models. In Frontiers in Pharmacology, 2020.

---

> ### Author Rebuttal · Authors · 2024-05-31
>
> Thanks for your time and feedback!
>
> **Q1. These results on the GuacaMol benchmarks are absent from the subsequent sections.**
>
> **A1.** Thanks for your comments! We believe there are misunderstandings. The results of unconditional molecule generation with GuacaMol benchmarks are put in Table 5 of Appendix.
>
> **Q2. Why are the evaluation metrics from MolT5 missing?**
>
> **A2.** Thanks for your question! The metrics like BLEU, Exact, Levenshtein employed in MolT5 are used to assess similarity based on SMILES representations rather than graph similarity. **We have used MACCS similarity metrics in our paper.** Additionally, our paper conduct molecule generation with a focus not just on reproducing known molecules, but on generating diverse and novel molecules that have not yet been discovered, aiming to advance drug discovery.
>
> **Q3. The comparison should include baselines in [1]. Table 5 should be moved to the main body.**
>
> **A3.** Thanks for your comments! We have included the baseline from the source code provided in [1]. The results presented below demonstrate that 3M-Diffusion outperforms all baselines across all evaluated metrics.
>
> |   | ChEBI-20 |       |       |       |       | PubChem |       |       |       |
> |-----------|----------|-------|-------|-------|-------|---------|-------|-------|-------|
> |           | Uniq (%) | Novelty (%) | KL Div (%) | FCD (%) | Validity (%) | Uniq (%) | Novelty (%) | KL Div (%) | FCD (%) | Validity (%) |
> | CharRNN   | 72.46    | 11.57  | 95.21 | 75.95 | 98.21 | 63.28   | 23.47  | 90.72 | 76.02 | 94.09      |
> | VAE       | 57.57    | 47.88  | 95.47 | 74.19 | 63.84 | 44.45   | 42.47  | 91.67 | 55.56 | 94.10      |
> | AAE       | 1.23     | 1.23   | 38.47 | 0.06  | 1.35  | 2.94    | 3.21   | 39.33 | 0.08  | 1.97       |
> | LatentGAN | 66.93    | 57.52  | 94.38 | 76.65 | 73.02 | 52.00   | 50.36  | 91.38 | 57.38 | 53.62      |
> | 3M-Diffusion | **83.04** | **70.80** | **96.29** | **77.83** | **100.00** | **85.42** | **81.20** | **92.67** | **58.27** | **100.00** |
>
>
> Our paper mainly focuses on text-guided molecule generation. While unconditional generation isn't our main focus, it serves to validate our model's effectiveness in learning diverse and novel molecules. Therefore, we include the results of unconditional molecule generation in the Appendix rather than in the main body.
>
>
> **Q4. Table 1 and Table 2 have the same caption.**
>
> **A4.** Thanks for your comments! We will correct this problem in our revised version.

---

> > ### Author Response · Authors · 2024-06-03
> > **A sincere and kind reminder to the reviewer**
> >
> > Dear Reviewer xhrH:
> >
> > We gratefully appreciate your time in reviewing our paper and your comments.
> >
> > We have made extensive efforts to address your comments and believe that they adequately address all your concerns. The reviewer's comments are mainly about some clarifications and are not fatal to the contributions of our paper; we believe that the reviewer's insightful comments can be easily and effectively addressed in the final version.
> >
> > With only four days left in the discussion period, we would like to confirm whether there are any other clarifications they would like. If the reviewer's concerns are clarified, we would be grateful if the reviewer could increase the score.
> >
> > Many thanks for your time; we are extremely grateful.
> >
> > The authors of "3M-Diffusion: Latent Multi-Modal Diffusion for Language-Guided Molecule Generation"

---

> > > ### Author Response · Authors · 2024-06-05
> > > **Kind reminder to reviewer xhrH: the author-reviewer discussion period is coming to an end.**
> > >
> > > Dear Reviewer xhrH,
> > >
> > > We want to sincerely thank you again for your time and comments!
> > >
> > > We have made extensive efforts to address your comments and carefully considered your suggestions point-by-point and added unconditional molecule generation you request in the rebuttal. We would like to kindly remind you that we are approaching the end of the author-reviewer discussion. In light of our rebuttal, we kindly ask if you could consider increasing your score. Thank you very much once again, and we look forward to hearing back from you if you have further comments.
> > >
> > > Thank you very much for your time.
> > >
> > > Best regards,
> > >
> > > The Authors

---

### Official Review · Reviewer_c27h · 2024-05-10

**Rating:** 6
**Confidence:** 3
**Ethics Flag:** 1

**Summary:**

The paper introduces a diffusion model that generates molecular graphs using textual descriptions of molecules as conditions. This framework first aligns the representations of molecular graphs and text descriptions. It then proceeds to train in two phases: an encoder-decoder and a multi-modal latent diffusion model.

**Reasons To Accept:**

- The paper proposes a novel method called 3M-Diffusion for generating diverse and novel molecular graphs from textual descriptions. The multi-modal diffusion approach that operates on an aligned latent space of graphs and text appears innovative and well-motivated to address limitations of prior work.
- The experiments on multiple real-world datasets demonstrate that 3M-Diffusion outperforms state-of-the-art baselines, especially in terms of the novelty and diversity of generated molecules, while still maintaining semantic consistency with the text prompts. For example, it achieves over 146% improvement in novelty and 130% in diversity on the PCDes dataset.

**Reasons To Reject:**

- Figure 1 is confusing; the encoder and condition should be inputted together into the diffusion model.
- The legends in Table 1 are incorrect for the datasets ChEBI-20 and PubChem.
- Tables 1 and 2 could be merged to reduce redundancy.
- How to obtain the ground truth graph of a generated molecule when calculating similarity if the descriptions of the two texts are very similar.
Additional experiments, such as retrieval tasks, should be conducted to validate the alignment of representations between text and molecular graphs.
- Does the case study use examples taken from the training set?
- It is inappropriate to express improvements in novelty and diversity as the percentage increases. The statement '3M-Diffusion achieves 146.27% novelty and 130.04% diversity relative improvement over the SOTA method' should be rephrased to better reflect these metrics.
- Due to the lack of distinctiveness in the similarity, novelty, diversity, and validity metrics, it would be beneficial to identify additional indicators for assessing the algorithm, such as plotting the distribution of logP and QED.
- Why is there such a significant difference between the prompts in Figure 2 and Figure 3? The prompt in Figure 3 includes the molecular components, whereas Figure 2 only provides a simple description of properties. The disparity is too significant.
The paper's statement, 'For example, the structure of ethanol can be specified using CCO, OCC, or C(O)C,' requires supporting evidence. If the method for generating SMILES is consistent, would this issue not exist?

---

> ### Author Rebuttal · Authors · 2024-05-31
>
> We thank the reviewer for the comments.
>
> **Q1. Figure 1 is confusing, the legends in Table 1 are incorrect and Table 1 and 2 could be merged.**
>
> **A1.** Thanks for your comments! We will update the Figure and Tables following your suggestion.
>
>
> **Q2. How to obtain the ground truth graph of a generated molecule if two texts are very similar.**
>
> **A2.** We believe there may be some misunderstandings. Each sample in the dataset consists of a text paired with a **molecular ground truth graph (included in the dataset)**.
>
> **Q3. Retrieval tasks should be included.**
>
> **A3.** We present the molecule-to-text retrieval results on PubChem, comparing them with Sci-BERT:
>
>
> |         | Acc  | R@20  |
> |------------|------|-------|
> | Sci-BERT   | 39.7 | 85.8  |
> | 3M-Diffusion   | **61.7** | **90.8**  |
>
> We measure performance with accuracy and recall@20, where our model outperforms Sci-BERT.
>
> **Q4. Does the case study from the training set?**
>
> **A4.** No, we select the samples from the test set.
>
> **Q5. It is inappropriate to express improvements as the percentage increases.**
>
> **A5.** Thank you for your comments. We will use absolute relative improvement.
>
> **Q6. Plot the distribution of logP and QED.**
>
> **A6.** Thanks for your comments! We have plotted the logP and QED distribution on the ChEBI-20 dataset for ground truth data, MolT5-large and 3M-Diffusion in the [link](https://anonymous.4open.science/r/3M-Diffusion-2217/distribution.pdf). Our model shows smaller or similar distribution differences with the ground truth compared to MolT5-large, highlighting its effectiveness.
>
> **Q7. Why is there such a big difference between the prompts in Figure 2 and 3?**
>
> **A7.** Thanks for your questions! The text in Figure 3 is derived from the original dataset, and the property-related words used in Figure 2 is designed by ourselves. We try to explore our model's ability for text-guided molecule generation of different cases with different prompts.
>
> **Q8. The paper's statement requires supporting evidence. If the method generate consistent SMILES, would this issue not exist?**
>
> **A8.** Thanks for your question! The structure of ethanol can be specified using CCO, OCC, or C(O)C,' is in the [link](https://en.wikipedia.org/wiki/Simplified_molecular-input_line-entry_system). While the model delivers consistent results, variations in SMILES can yield molecules with distinct properties, as noted in JT-VAE, making it also unsuitable to use SMILES for molecule representation.

---

> > ### Author Response · Authors · 2024-05-31
> > **Additional Experiment Results**
> >
> > Thanks again for your effort on reviewing our paper. Due to the word limit in the previous rebuttal part, we provide the following additional demonstrations and experiments by your suggestions:
> >
> > **Additional experiment and demonstrations for Q3**
> >
> > We present molecule-to-text retrieval (M2T) and text-to-molecule retrieval (T2M) tasks by comparing with Sci-BERT [1] and KV-PLM [2] on PubChem. Sci-BERT, pretrained on scientific texts, serves as the initial text encoder for the 3M-Diffusion model. Similarly, KV-PLM, a deep learning system, can be utilized for molecule and text retrieval tasks. Both models employ SMILES to facilitate the molecule and text tasks. Here are the following results:
> >
> > | Model      | M2T  |       | T2M  |       |
> > |------------|------|-------|------|-------|
> > |            | Acc  | R@20  | Acc  | R@20  |
> > | Sci-BERT   | 39.7 | 85.8  | 37.5 | 85.2  |
> > | KV-PLM     | 38.8 | 86.0  | 37.7 | 85.5  |
> > | 3M-Diffusion     | **61.7** | **90.8**  | **63.2** | **90.2**  |
> >
> > We observe that 3M-Diffusion can outperform Sci-BERT and KV-PLM, which verifies the effectiveness of our model for the retrieval task.
> >
> > [1] SciBERT: A Pretrained Language Model for Scientific Text.
> >
> > [2] A deep-learning system bridging molecule structure and biomedical text with comprehension comparable to human professionals.

---

### Official Review · Reviewer_H3jC · 2024-05-11

**Rating:** 6
**Confidence:** 4
**Ethics Flag:** 1

**Summary:**

This paper studies language-guided molecule generation given text description. The proposed method consists of multiple neural networks, including a graph encoder, decoder, text encoder, and a diffusion model in the latent space. To align the representations of text and graphs in the latent space, the authors utilized contrastive loss. Experiments on conditional and unconditional molecule generation were conducted to demonstrate the effectiveness of the proposed framework.

**Reasons To Accept:**

1. **Less explored yet important topic.** The molecule generation using language descriptions is relatively less explored topic and it has many practical applications such as drug design and material discovery. In addition, the graph generation is relatively challenging since graphs are irregular without clear directions and they have a various number of nodes and edges.
2. **Ablation study.** The authors provide the ablation study to show the contribution of each component, including two-stage training and representation alignment.

**Reasons To Reject:**

1. **Overclaim and SOTA.** The key contribution, (iii) the proposed method outperforms SOTA methods, is questionable. First of all, the baselines did not include recent methods, and unconditional generation results are not provided in the table.
2. **No technical contribution.** The proposed pipeline is a combination of existing modules. HierVAE, Latent diffusion, GIN for graph encoder and contrastive loss for text and graph latent space alignment. Also, no significant modification/adaption of existing methods for text-guided molecule generation.
3. **Trade-off rather than improvement.** Table 1 and Table 2 show that the proposed method shows a noticeable degradation in similarity and, instead, performance gains in other metrics. This could be viewed as trade-off between metrics. In addition, Novelty, Diversity and Validity are not related to language-guidance. They are not specifically designed to evaluate the fidelity between conditions and generated samples.

---

> ### Author Rebuttal · Authors · 2024-05-31
>
> We gratefully appreciate your time in reviewing our paper.
>
> **Q1. Overclaim and SOTA.**
>
> **A1.** Thanks for your comments. We add Mol-instruct (ICLR 2024) [1] as a baseline. The results show that 3M-Diffusion outperforms Mol-instruct (Due to word limits, we just put results on PCDs, all other datasets have similar conclusions).
>
> |          |   | Similarity (%) | Novelty (%) | Diversity (%) | Validity (%) |
> |-----------|---|------|--------|--------|---------|
> | Mol-instruct | | 60.86 | 35.60 | 24.57 | 79.19 |
> | 3M-Diffusion  | | **81.57** | **63.66** | **32.39** | **100.0** |
>
> For the unconditional generation results, we put them in Table 5 of the Appendix.
>
> **Q2. No technical contribution.**
>
> **A2.** We respectfully disagree that our model isn't novel or has no technical contributions because it is not surprising, and uses components in previous works. We believe that discouraging authors from building on others' work due to fear of reviewer criticism hinders community progress. We argue that we explore text-guided molecule generation from a novel perspective that involves generating molecular graphs based on texts with diffusion models.
>
> While the diffusion model has shown promise in various fields, it has not yet been applied to the novel problem of text-guided molecular graphs. We stress the fact that our method can be used to discover novel, diverse and high-quality molecules.
>
> **Q3. Trade-off rather than improvement.**
>
> **A3.** We respectfully disagree that our model's performance is merely a trade-off rather than an improvement. Trade-offs are inherent in molecule generation. Generating molecules that match text is important, but creating diverse and novel molecules is also crucial in drug discovery. A model that only reconstructs known molecules but cannot discover new ones fails to meet drug discovery needs. Therefore, the target of text-guided molecule generation maintains an acceptable level of similarity while ensuring diversity and novelty. Overall, our average results on all metrics outperform other methods as shown below (just put PCDes for world limit and all datasets have similar conclusion) :
>
> |         | Avg |
> |---------------|--------|
> | MolT5-small   | 50.06    |
> | MolT5-base    | 53.96    |
> | MolT5-large   | 54.77    |
> | ChemT5-small  | 55.1    |
> | ChemT5-base   | 55.42    |
> | 3M-Diffusion   | **69.12**    |
>
> [1]Mol-instructions: A large-scale biomolecular instruction dataset for large language models.

---

> > ### Author Response · Authors · 2024-05-31
> > **Additional Experiment Results**
> >
> > Thank you once again for your comments and suggestions! Due to word limits of previous rebuttal part, we have provided additional results of more datasets for the experiments you suggested in this part.
> >
> > **Additional Experiments for Q1 on PCDes, Momu, ChEBI-20 and PubChem datasets:**
> >
> > |   |  | PCDes |        |        |         | MoMu |       |        |         |
> > |-----------|---|------|--------|--------|---------|------|-------|--------|---------|
> > |     Metrics      |   | Similarity (%) | Novelty (%) | Diversity (%) | Validity (%) | Similarity (%) | Novelty (%) | Diversity (%) | Validity (%) |
> > | Mol-instruct | | 60.86 | 35.60 | 24.57 | 79.19 | 14.89 | 97.52 | 30.17 | 68.32    |
> > | 3M-Diffusion  | | **81.57** | **63.66** | **32.39** | **100.0** | **24.62** | **98.16** | **37.65** | **100.0**     |
> >
> > |   | ChEBI-20 |  |        |        |   PubChem  |   |       |        |         |
> > |-----------|---|------|--------|--------|---------|------|-------|--------|---------|
> > | Metrics         | Similarity (%) | Novelty (%) | Diversity (%) | Validity (%) | Similarity (%) | Novelty (%) | Diversity (%) | Validity (%) |
> > | Mol-Instruct | 65.75         | 32.01       | 26.50         | 77.91        | 23.40          | 37.37        | 27.97         | 71.10        |
> > | 3M-Diffusion  | **87.09**         |  **55.36**       | **34.03**         | **100**          | **87.05**          | **64.41**        | **33.44**         | **100**          |
> >
> > **Additional Experiments for Q3  on all datasets (average results of all metrics are put in the following table):**
> >
> > | Model         | PCDes | MoMu | ChEBI-20 | PubChem |
> > |---------------|--------|--------|--------|--------|
> > | MolT5-small   | 50.06    | 51.06 | 50.06 | 44.29 |
> > | MolT5-base    | 53.96    | 46.37 | 53.97 | 46.37 |
> > | MolT5-large   | 54.77    | 54.36 | 54.36 | 54.36 |
> > | ChemT5-small  | 55.10    | 54.20 | 55.10 | 54.20 |
> > | ChemT5-base   | 55.42    | 54.56 | 55.42 | 54.56 |
> > |    3M-Diffusion  | **69.12** | **65.10** | **69.12** |  **71.23** |
> >
> > **We can find that 3M-diffusion's average results on all metrics outperform other baselines, which show the improvement of our model.**

---

> > > ### Comment · Reviewer_H3jC · 2024-06-03
> > >
> > > I appreciate the authors' detailed responses. As the experimental results, the proposed method shows strong performance. The response to the technical novelty is not convincing and novelty cannot be addressed by the response unless there exists any misunderstanding. Although this paper has limited technical contributions, this is an early attempt for molecule generation, showing great performance. I raise my rating.

---

> > > > ### Author Response · Authors · 2024-06-03
> > > > **Thank you for your response to our rebuttal**
> > > >
> > > > Dear Reviewer H3jC:
> > > >
> > > > Thank you very much for reviewing our paper and reading our rebuttal. We sincerely appreciate your recognition of our clarifications and the increase in your score!
> > > >
> > > > **Thank you also for your additional comments to facilitate further discussion, which will further improve our paper.**
> > > >
> > > > **In the responses below, we have carefully addressed your remaining concerns about technical novelty.**
> > > >
> > > > We are truly grateful for your time and your reply.
> > > >
> > > > Best regards, Authors

---

> > > > > ### Author Response · Authors · 2024-06-03
> > > > > **Further Responses and Clarifications to Reviewer H3jC**
> > > > >
> > > > > We sincerely thank the reviewer for the additional comments and suggestions. Please see our detailed clarifications about technical novelty below:
> > > > >
> > > > > - 3M-Diffusion studies a novel problem about text-guided molecular graph generations. **The task of generating diverse, novel molecules that match textual descriptions is challenging due to the sparsity and discrete nature of graph structures.** While there is extensive research on using variational autoencoders (VAEs) or diffusion models (DMs) for molecular graph generation to produce diverse, novel or property-related molecules, there is no existing work that leverages the capabilities of VAEs or DMs to generate molecules that are both diverse, novel, and text-matching. **We believe that the 3M-Diffusion model makes significant technical contributions and novelty by establishing a pipeline capable of generating diverse, novel and text-matching molecular graphs.** This pipeline achieves performance comparable to transformer-based models in terms of "Similarity," which measures the model's text-matching capabilities, and excels at generating diverse and novel molecules.
> > > > >
> > > > > - Furthermore, we are the first to explore the ability of diffusion models on the latent space of molecular graphs to generate novel, diverse and text-matching molecules. **Specifically, our technical contributions and novel approaches include designing the latent space and creating a suitable scoring networks for latent diffusion tailored to the latent space of molecular graphs, enabling our model to function effectively.**
> > > > >
> > > > > - Also, in our 3M-Diffusion model, **we do not directly utilize HierVAE's encoder; instead, we employ our own trained encoder that aligns graph and text representations, while only incorporating HierVAE's decoder.** Unlike HierVAE's encoder, which learns hierarchical graph representations but can’t understand text, we propose using an aligned encoder from GIN. This aligned encoder is designed to understand textual descriptions by contrastive learning. **Utilizing this encoder's representation for the decoder and making it effectively generate diverse, novel, and text-matching molecules, represents a significant technical novelty and contribution of our model.**

---

### Official Review · Reviewer_8fJY · 2024-05-11

**Rating:** 6
**Confidence:** 3
**Ethics Flag:** 1

**Summary:**

This paper proposes a language model on molecule generation, well combining the existing practices in multi-modal generation: (1) train a VAE model for molecule graph generation and text descriptor generation; (2) align the representation of molecule graph and text descriptors; (3) introduce a latent diffusion model for mapping the text spaces to the latent molecular graph spaces, which then can be decoded to a molecular graph.
Experiment results show the proposed method is effective, outperforming SOTA methods for both text-guided and unconditional molecule generation.

**Questions To Authors:**

1. How about using the 3D geometry for molecular modeling and introducing the protein or pocket information as a prompt for drug discovery?

**Reasons To Accept:**

1. Effectiveness: The proposed method is effective and shows strong performances in both text-guided and unconditional molecule generation.
2. Well-motivated: Introducing molecular graph modeling in text-guided molecule generation is well-motivated.
3. Well writing.

**Reasons To Reject:**

1. Aligning the concept of molecular to text generation is useful, which will help us to understand the properties of unknown or new molecules. In such a case, it is more natural for me to understand the necessity of molecule-to-text generation instead of text-guided molecular generation.
2. The technical novelty is limited, but it is acceptable.

---

> ### Author Rebuttal · Authors · 2024-05-30
>
> We appreciate the reviewer's perception of our contributions to both empirical analysis and motivations, and are grateful for the encouraging comments. Please see our responses below:
>
> **Q1. It is more natural for me to understand the necessity of molecule-to-text generation instead of text-guided molecular generation.**
>
> **A1.** Thank you for your comments. Traditional Molecule generation based on value guidance specifying a single or a handful of target properties might be insufficient to capture intricate conditions. Using natural language can greatly utilize biomedical knowledge from humans for molecule generation [1], [2]. Specifically, text-guided molecule generation allows us to encompass conditions adeptly and flexibly, like substructure (benzene ring) or the novel properties like categories of the molecule (“flavones” as shown in Figure 2).
>
>
> **Q2. The technical novelty is limited, but it is acceptable.**
>
> **A2.** Thank you for your feedback on the technical novelty of our work! We believe that our approach introduces several innovative aspects. Specifically, our paper highlights:
>
> - **novel problem**: we explore the previously unaddressed problem of text-guided molecular graph generation.
>
> - **novel perspective**: we present an approach from a novel perspective based on diffusion models for the text-guided molecular graph generation.
>
> - **novel results**: our 3M-Diffusion model successfully generates diverse and high-quality molecules that accurately match textual descriptions, which outperform other baselines for both text-guided and unconditional molecule generation.
>
>
> **Q3. How about using the 3D geometry for molecular modeling and introducing the protein or pocket information as a prompt for drug discovery?**
>
> **A3.** Thanks for your questions! We agree with your opinions. However, our current paper is focused on text-guided molecule generation and this concept falls outside our scope. We acknowledge the potential of this approach and suggest exploring these promising settings in future work.
>
> [1] Edwards, Carl, et al. "Translation between Molecules and Natural Language." EMNLP-2022.
>
> [2] Christofidellis, Dimitrios, et al. "Unifying molecular and textual representations via multi-task language modeling." ICML-2023.

---

### Author Response · Authors · 2024-06-05
**General Response by Authors**

Dear Reviewers,

We sincerely thank all the reviewers for their thoughtful feedback and their time. We are encouraged to see all reviewers agree that our paper is a novel, interesting, and effective work with strong experimental results.

In our work, we propose the first multimodal diffusion approach to generating molecular graphs from their textual descriptions, which is critical for molecule generation or drug discovery with textual guidance. Our proposed 3M-Diffusion effectively aligns the latent spaces of molecular graphs and textual descriptions to offer a text-molecule aligned latent diffusion model to generate high-quality, diverse, and novel molecular graphs that match the textual description provided. We demonstrate that our 3M-Diffusion has great empirical performance on both text-guided molecule generation and unconditional molecule generation compared with baselines. These empirical results and a novel perspective for text-guided molecule generation were found to be particularly interesting by all reviewers.

We have made our greatest efforts to incorporate all questions, suggestions, and comments in our response to each reviewer, and have prepared a point-by-point response for each. Briefly, we have addressed key themes in the reviews, including additional comparisons with baselines, retrieval tasks, misunderstandings, and minor questions. Below, we summarize the major responses, while we address the comments of each reviewer separately.

- We added the additional comparisons with baselines for text-guided molecule generation and unconditional generation (Reviewer H3jC, xhrH).
- We added retrieval tasks for our text and graph encoder (Reviewer c27h).
- We added additional clarification about technical novelty and contributions (Reviewer H3jC, 8fJY).
- We compared 3M-Diffusion with the baseline by plotting the distribution of logP and QED. (Reviewer c27h).

For the other questions, we have provided individual responses to each one. We will update the paper to include these additional experiments and clarifications in response to your comments.

We extend our sincere thanks to all the reviewers once again for their time and efforts.

Best regards,

Authors

---

### Decision · Program_Chairs · 2024-07-10

**Decision:**

Accept

**Comment:**

This work presents a method to guide molecular generation with NL. The key concern being improving on SOA through improving the diversity of molecules generated. the work accomplishes its task, but the requires several different models, each doing a specific task to be trained together. This is is unsurprising because the work builds on pretrained models (graph/llm) which raises the question of what becomes more efficient if you build end-to-end.

As reviewer H3jC points out, the changes wrt SOA methods need to be evidenced/explained more, especially given the substantial change in the fidelity of the model (c.f. Mol-instruct numbers shared in the rebuttal). These big changes should be explained in terms of what this model is fixing/ignoring wrt the baseline, for example are there particular types of structure/strategies that increase the diversity for 3M compared to Mol-instruct?  Why is Validity always 100% for 3M, are there cases where that is not the case? Have you done any human eval on a small subset to understand how well aligned generated molecules are with descriptions?

Overall, this is a nice work and the thorough baselines that are useful to the community. I would encourage authors to share as many artifacts from data to models as possible